

# Investigation of Helicobacter *pylori* infection among symptomatic children in Hangzhou from 2007 to 2014: a retrospective study with 12,796 cases

Xiaoli Shu[1], Mingfang Ping[1,2], Guofeng Yin[1,3] and Mizu Jiang[1]

[1] Gastrointestinal Laboratory, the Children's Hospital, Zhejiang University School of Medicine, Hangzhou, China
[2] Current affiliation: Department of Pediatrics, Second Affiliated Hospital of Jiaxing University, Jiaxing, China
[3] Current affiliation: Department of Pediatrics, Shaoxing Women & Children's Hospital, Shaoxing, China

## ABSTRACT

**Background and Aim**. The infection of Helicobacter *pylori* (H. *pylori*) is acquired in childhood and the prevalence vary greatly in different countries and regions. The study aimed to investigate the characteristics of H. *pylori* infection among children with gastrointestinal symptoms in Hangzhou, a representative city of eastern China.

**Methods**. A systematic surveillance of H. *pylori* infection according to the $^{13}$C-urea breath test was conducted from January 2007 to December 2014 in the Children's hospital, Zhejiang University School of Medicine. The demographic information and main symptoms of every subject were recorded.

**Results**. A total of 12,796 subjects were recruited and 18.6% children evaluated as H. *pylori* positive. The annual positive rates decreased from 2007 to 2014 ($\chi^2 = 20.461, p < 0.01$). The positive rates were 14.8%, 20.2% and 25.8% in 3–6, 7–11 and 12–17 years age group respectively, which increased with age ($\chi^2 = 116.002, p < 0.01$). And it was significantly higher in boys than girls ($\chi^2 = 15.090, p < 0.01$). Multivariate logistic regression identified possible risk factors for H. *pylori* infection. Age, gender, gastrointestinal symptoms and history of H. *pylori* infected family member were all significantly associated with H. *pylori* infection (all $p < 0.05$).

**Conclusions**. H. *pylori* infection rates in children with gastrointestinal symptoms were lower than most of those reported in mainland China. Further studies are required to determine the prevalence in the general population. Comprehensively understanding of the characteristics and the possible risk factors of H. *pylori* infection will be helpful to its management strategies in children in China.

Corresponding author
Mizu Jiang, mizu@zju.edu.cn

## INTRODUCTION

Helicobacter *pylori* (H. *pylori*) is a Gram-negative, microaerophilic bacterium which selectively colonizes in the human stomach mucosa. The prevalence of H. *pylori* infection is about 50% of the world's population and gastric cancer related to H. *pylori* infection is the fourth most common cancer and the second leading cause of cancer-related death

worldwide (*Atherton & Blaser, 2009*). In general, the prevalence in less developed or developing countries is higher than that in developed countries (*Fock & Ang, 2010*). The infection rates are reported varying from 15.5% to 93.6% in developed and developing countries, respectively (*Eusebi, Zagari & Bazzoli, 2014*; *Mentis, Lehours & Mégraud, 2015*; *Tonkic et al., 2012*).

It is now accepted that H. *pylori* infection is acquired in childhood (*Rowland et al., 2006*), and H. *pylori* generally persists for the life of the host in the absence of antibiotic therapy (*Pacifico et al., 2010*). The incidence and prevalence rates of childhood infection with H. *pylori* also vary greatly worldwide. Within developed nations, prevalence rates of H. *pylori* infection among children have been shown to range from 6.5% to 65% (*Roma & Miele, 2015*; *Tonkic et al., 2012*). Now in European and North America, the epidemiology of H. *pylori* infection in children has changed in recent decades with low incidence rates, which resulting in prevalence lower than 10% in children and adolescents (*Kindermann & Lopes, 2009*). However, there were few reports in developing counties. There has been a decrease in the H. *pylori* infection rate in the general Chinese population in recent years but it also remained high in some areas among both children and adults after fifteen years (*Ding et al., 2015*; *Zhang et al., 2009a*).

China is regarded as one of the largest developing country inhabited by more than one-fifth of the world's population although there has been rapid growth in economy in the past decade. The very limited data showed that the prevalence rate of H. *pylori* infection in Chinese children ranged from 6.8% in three cities of China to 72.3% in northwest China with large regional variations (*Ding et al., 2015*; *Zhang et al., 2009b*). Hangzhou, the capital city of Zhejiang Province, which had made quick improvements in industrialization and socioeconomic conditions since the 1980s, is a representative city of eastern China. But few studies have assessed the prevalence of H. *pylori* infection in this area. The lack of these data in our pediatric population has hampered the better understanding of the disease burden in our society and the healthcare planning for resources allocation to tackle H. *pylori*-associated diseases which are usually encountered in adulthood. The aim of this study was to estimate the prevalence of H. *pylori* infection among children in Hangzhou, China from 2007 to 2014 and evaluate the characteristics of H. *pylori* infection in children.

## METHODS

### Study population

Subjects aged from three to 18 years old who were referred for the detection of H. *pylori* infection using $^{13}$C-urea breath test ($^{13}$C-UBT) were recruited at the Children's hospital, Zhejiang University School of Medicine from January 1, 2007 to December 31, 2014. The main symptoms of every subject, besides a history of H. *pylori* infected family member were recorded, including abdominal pain, anorexia, nausea/vomiting, abdominal distension, hiccup, constipation, halitosis, diarrhea and failure to thrive/weight loss. All children should have been fasting more than 6 h, and had not used bismuth salts, proton-pump inhibitors (PPIs), or any antibiotics (amoxicillin, tetracycline, metronidazole, clarithromycin, azithromycin, or other) within one month before the $^{13}$C-UBT (*Koletzko et al., 2011*).

The major exclusion criteria included: age younger than three or older than 18, children with incomplete patient data, patients who previously diagnosed as H. *pylori* infection and received treatment for H. *pylori* infection even with drug withdrawal 4 weeks prior to the $^{13}$C-UBT.

### Detection of H. *pylori* infection

H. *pylori* infection was established by the $^{13}$C-UBT kit, Helikit (Isodiagnostika Inc., Edmonton, AB, Canada) according to standard protocols. Briefly, after a minimum fasting period of 6 h, a baseline exhaled breath sample was obtained using a collection bag. The children then drank 75 ml of a citrus-flavoured liquid preparation (75 mg of $^{13}$C-labelled urea). Thirty minutes later, another breath exhaled sample was stored in collection bag. Breath samples were stored at room temperature and then analyzed by an isotope selective nondispersive infrared spectrometer, namely by ISOMAX 2000 (Isodiagnostika Inc., Edmonton, AB, Canada). The test was defined as positive when delta over baseline (DOB) value calculated after thirty minutes was 3.5 $\delta$‰ or more (*Mauro et al., 2006*).

### Statistics

Descriptive statistics such as median and interquartile range of age, percentages were calculated for demographic data and results were analyzed by chi-squaretest. The distribution of H. *pylori* infection rate by year was analyzed by Linear-by-Linear association. Multivariate logistic regression analysis was used to control for the potential confounding variables associated with H. *pylori* infection. Results of logistic regression were expressed as odds ratios (OR) with 95% confidence intervals (CI). Statistical analysis was performed using SPSS version 19.0 (SPSS Inc, USA) and $P$ value was calculated. Two tailed $P < 0.05$ was considered statistically significant.

### Ethical considerations

The study was approved by Institutional Review Board and Institutional Ethics Committees of the Children's hospital, Zhejiang University School of Medicine (2016-IRBAL-078).

## RESULTS

### Demographic data

A total of 12,796 subjects were enrolled in this study and there were 6,880 boys and 5,916 girls, yielding a male-to-female ratio of 1.16:1. All children were divided into three age groups, including 3–6 (pre-school age), 7–11 (school age) and 12–17 (adolescent) years age group. The gender distribution was consistent in different age groups. The median and interquartile range of age of all children were 7.50 (5.75–10.08) years, while boys were 7.50 (5.67–10.08) years and girls were 7.58 (5.83–10.08) years.

### H. *pylori* infection rate

Overall, 18.6% (2,382/12,796) children were H. *pylori* positive according to the DOB value of 13C-UBT (Table 1). The annual positive rates decreased from 2007 to 2014 ($\chi^2 = 20.461$, $p < 0.01$) (Fig. 1). And the infection rate decreased in the latest four-year period 2011–2014, compared to the former four-year period 2007–2010 ($\chi^2 = 25.798$, $p < 0.01$) (Fig. 2). The

**Table 1  Demographic characteristics of the 12,796 subjects.**

| | H. *pylori*-positive | H. *pylori*-negative | Total | *P* value |
|---|---|---|---|---|
| **Age groups (years)** | | | | |
| 3–6 | 800 (14.8) | 4,608 (85.2) | 5,408 | <0.001 |
| 7–11 | 1,179 (20.2) | 4,650 (79.8) | 5,829 | |
| 12–17 | 403 (25.8) | 1,156 (74.2) | 1,559 | |
| **Gender** | | | | |
| Female | 1,016 (17.2) | 4,900 (82.8) | 5,916 | <0.001 |
| Male | 1,366 (19.9) | 5,514 (80.1) | 6,880 | |
| **Gastrointestinal symptoms** | | | | |
| No | 432 (17.5) | 2,034 (82.5) | 2,466 | 0.119 |
| Yes | 1,950 (18.9) | 8,380 (81.1) | 10,330 | |
| **History of H. *pylori* infected family member** | | | | |
| No | 2,139 (18.4) | 9,488 (81.6) | 11,627 | 0.045 |
| Yes | 243 (20.8) | 926 (79.2) | 1,169 | |
| **Total** | 2,382 (18.6) | 10,414 (81.4) | 12,796 | – |

Notes.
Data expressed as number (%).

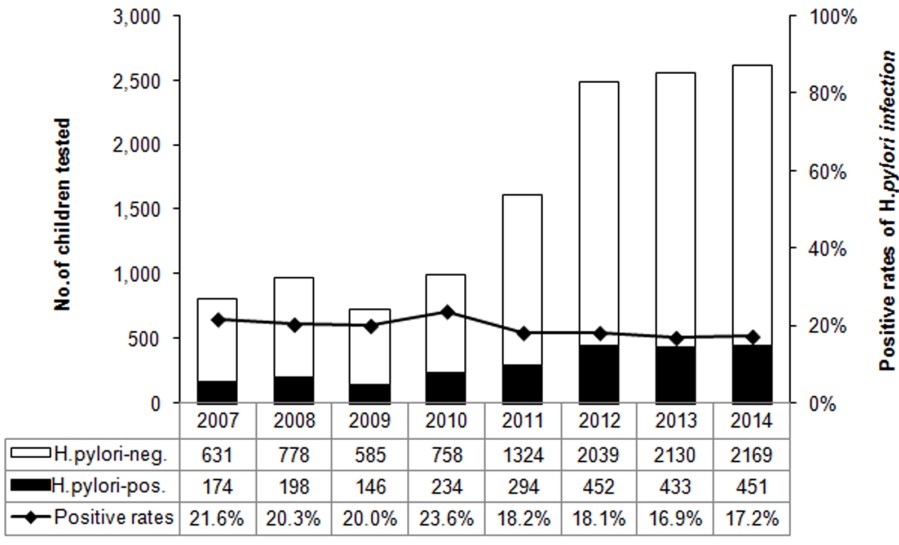

**Figure 1  The distribution of H. *pylori* infection rate by year from 2007 to 2014.** The bars represent the number of enrolled subjects each year. H. *pylori* negative and positive subjects are white and black respectively. The line chart represent the positive rates of H. *pylori* infection each year.

positive rates of H. *pylori* was 14.8% (800/5,408) in 3–6 years age group, 20.2% (1,179/5,829) in 7–11 years age group, and 25.8% (403/1,559) in 12–17 years age group, which increased with age and were statistically significant ($\chi^2 = 116.002, p < 0.001$) (Table 1). Furthermore, the positive rates were higher in boys (19.9%, 1,366/6,880) than girls (17.2%, 1,016/5,916), and the difference was also statistically significant ($\chi^2 = 15.090, p < 0.001$) (Table 1).
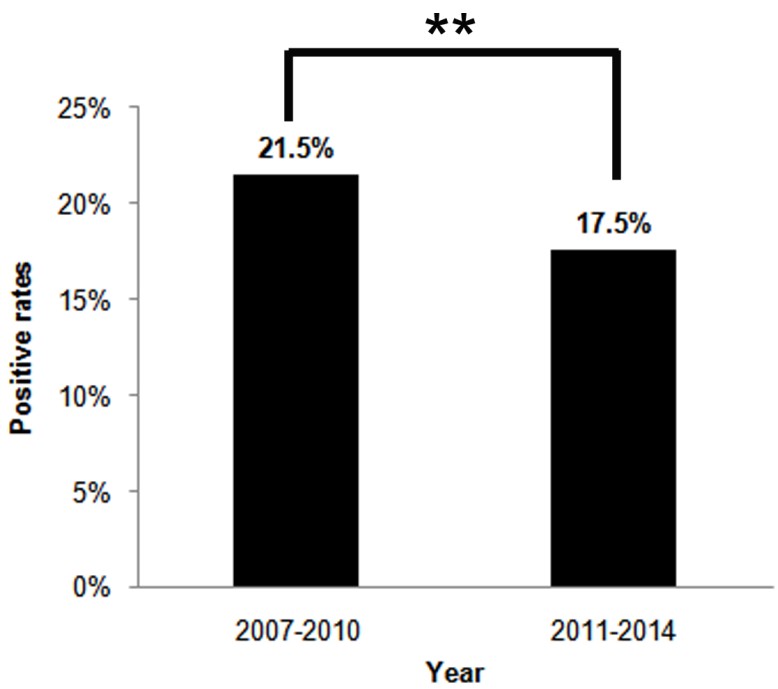

**Figure 2** **The H. *pylori* infection rates between two four-year period, 2007–2010 and 2011–2014.** The percentages on top of the bars represent the total H. *pylori* infection rates in four-year periods. **$p < 0.01$.

The main gastrointestinal symptoms of children undergoing $^{13}$C-UBT are abdominal pain, anorexia, nausea/vomiting, abdominal distension, hiccup, constipation, halitosis, diarrhea and failure to thrive/weight loss. There were 80.7% children (10,330/12,796) with at least one gastrointestinal symptom in the prior months. The positive rate of H. *pylori* infection in children with these symptoms was 18.9% (1,950/10,330), demonstrating no significant difference compared to 19.3% (2,466/12,796) children without gastrointestinal symptoms (17.5%, 432/2,466) ($\chi^2 = 2.426$, $p = 0.119$) (Table 1).

There were 1,169 children had a history of H. *pylori* infected family member, and the H. *pylori* infection rate was higher than those without a familial history (20.8% versus 18.4%, $\chi^2 = 4.005$, $p < 0.05$) (Table 1).

## Possible risk factors associated with H. *pylori* infection

Table 2 shows the results from the multivariate logistic regression performed to assess risk factors for H. *pylori* infection. Age, gender, gastrointestinal symptoms and history of H. *pylori* infected family member were found together to be significantly associated with H. *pylori* infection (all $p < 0.05$). Specifically, children in 7–11 years age group and in 12–17 years age group were 1.474 and 2.031 times as likely to be H. *pylori* infected as children in 3–6 years age group (95% CI [1.335–1.627] and 95% CI [1.772–2.328] respectively, all $p < 0.001$). Boys were 1.209 times as likely to be H. *pylori* infected as girls (95% CI [1.104–1.323], $p < 0.001$) and children with a history of H. *pylori* infected family member were 1.289 times compared to those without the familial history. Furthermore, gastrointestinal symptom was also one of risk factors for H. *pylori* infection, as it was 1.141

**Table 2** Logistic regression analysis for possible risk factors associated with H. *pylori* infection.

| Variables | OR (95%CI) | P value |
|---|---|---|
| **Age groups (years)** | | |
| 3–6 | – | |
| 7–11 | 1.474 (1.335–1.627) | <0.001 |
| 12–17 | 2.031 (1.772–2.328) | <0.001 |
| **Gender** | | |
| Female | – | |
| Male | 1.209 (1.104–1.323) | <0.001 |
| **Gastrointestinal symptoms** | | |
| No | – | |
| Yes | 1.141 (1.009–1.289) | 0.035 |
| **History of H. *pylori* infected family member** | | |
| No | – | |
| Yes | 1.289 (1.100–1.511) | 0.002 |

**Notes.**

OR, odds ratio; CI, confidence interval.

times in children with gastrointestinal symptoms compared to children without them (95% CI [1.009–1.289], $p < 0.05$).

# DISCUSSIONS

The present study assessed the $^{13}$C-UBT in the pre-treatment phase to evaluate current H. *pylori* infection in children with gastrointestinal symptoms. The prevalence was higher than in developed countries but lower than in some developing countries (*Tonkic et al., 2012*). It was higher than it reported in three cities (Beijing, Guangzhou and Chengdu) of mainland China, Hong Kong and Taiwan among asymptomatic children or school children, but lower than most of mainland China (Table 3). These could be due to cohort selection, detection method and the geographic area difference which may also reflect the personal and environmental hygiene. Subjects in our study enrolled from patients most of that had gastrointestinal symptoms and were suggested to detect the H. *pylori* infection, so the incidence rate would be more or less higher than asymptomatic or general population. Currently, there are many diagnostic tools to detect H. *pylori* infection, with non-invasive methods being considered as the most desirable for use especially in children. The $^{13}$C-UBT has been reported to have excellent sensitivity and specificity for the noninvasive identification of H. *pylori* infection in children and it is recommended for situations when endoscopy is not available or necessary (*Guarner et al., 2010*; *Redéen et al., 2011*). $^{13}$C-UBT has superiority over serologic methods by its high reliability and the ability to differentiate present from past infection (*Bourke et al., 2005*). The geographic distribution of H. *pylori* infection is correlated with the geographic distribution of gastric cancer. Muping County in Shandong Province, Wuwei County in Gansu Province and Jiangsu Province are all the area with high risk of gastric cancer (*Shi et al., 2008*; *Zhang et al., 2009a*; *Zhang et al., 2009b*). That may be associated with the high prevalence of H. *pylori* infection in this area.

**Table 3  Comparison of prevalence of H. *pylori* infection among children in China.**

| Authors | Recruitment | Area | Year | Age (year) | Method | No. | Prevalence (%) |
|---|---|---|---|---|---|---|---|
| Ding et al. (2015) | Asymptomatic children | Beijing | 2009–2011 | Newborn | HpSA | 330 | 0.6 |
| | | Guangzhou | | 1–12m | | 319 | 2.5 |
| | | Chengdu | | 1–3 | | 289 | 2.1 |
| | | | | 4–6 | | 624 | 7.2 |
| | | | | 7–9 | | 528 | 6.1 |
| | | | | 10–12 | | 308 | 11.0 |
| | | | | 13–15 | | 685 | 8.0 |
| | | | | 16–18 | | 408 | 13.5 |
| Tam et al. (2008) | School children | Hong Kong | 2007 | 6–8 | UBT | 300 | 9.3 |
| | | | | 9–10 | | 301 | 11.0 |
| | | | | 11–12 | | 472 | 14.8 |
| | | | | 13–14 | | 779 | 13.0 |
| | | | | 15–16 | | 289 | 12.5 |
| | | | | 17–19 | | 339 | 16.5 |
| Lin et al. (2007) | School children | Taiwan | 2004 | 9–12 | Serology | 1,625 | 11.0 |
| | | | | 13–15 | | 325 | 12.3 |
| Zhang et al. (2009a) | School children | Muping, Shandong | 2006 | 8–9 | HpSA | 122 | 26.2 |
| | | | | 10–11 | | 125 | 40.0 |
| | | | | 12–13 | | 142 | 41.6 |
| | | | | 14–15 | | 131 | 42.0 |
| | | Yanqing, Beijing | 2006 | 8–9 | HpSA | 130 | 15.4 |
| | | | | 10–11 | | 136 | 27.9 |
| | | | | 12–13 | | 125 | 29.6 |
| | | | | 14–15 | | 125 | 29.6 |
| Chen et al. (2007) | Population-based cohort | Guangzhou Guangdong | 2003 | 3–5 | Serology | 180 | 19.4 |
| | | | | 5–10 | | 105 | 22.9 |
| | | | | 10–20 | | 185 | 36.8 |
| Cheng et al. (2009) | Population-based cohort | Beijing | 2003 | 2–10 | UBT | 19 | 57.8 |
| | | | | 11–20 | | 52 | 46.2 |
| Shi et al. (2008) | Population-based cohort | Jiangsu | 2004–2005 | <20 | UBT/Serology | 48 | 60.4 |
| Zhang et al. (2009b) | Population-based cohort | Wuwei, Gansu | 2007–2008 | 3–5 | HpSA | 99 | 68.7 |
| | | | | 6–9 | | 240 | 70.4 |
| | | | | 10–14 | | 440 | 73.0 |
| | | | | 15–18 | | 159 | 75.5 |
| Zhang & Li (2012) | Gastrointestinal symptoms | Dongguan, Guangdong | 2010–2011 | 3–7 | Histology/ RUT/ UBT | 119 | 39.5 |
| | | | | 8–12 | | 134 | 41.0 |
| | | | | 13–16 | | 123 | 54.5 |
| Wu et al. (2008) | Gastrointestinal symptoms | Zunyi | 2000–2006 | 10–20 | UBT | 2,645 | 40.0 |
| Our study | Gastrointestinal symptoms | Hangzhou, Zhejiang | 2007–2014 | 3–6 | UBT | 5,408 | 14.8 |
| | | | | 7–11 | | 5,829 | 20.2 |
| | | | | 12–17 | | 1,559 | 25.8 |

**Notes.**
HpSA, H. *pylori* stool antigen test; UBT, urea breath test; RUT, rapid urease test; m, months.

Although there is apparent variation in the prevalence of H. *pylori* infection between developing and developed countries in children, it is reported all around the world that the prevalence was associated with age (*Tkachenko et al., 2007*; *Zhang et al., 2009a*). In our study, the prevalence of H. *pylori* infection was also shown to increase with age. Pre-school age children had a lower significant prevalence than school age and adolescent. The increase in H. *pylori* prevalence with age is thought to represent the improvements in socioeconomic conditions and sanitary standards through the generations. In Russia, the prevalence of H. *pylori* infection reduced markedly within a 10-year period (from 1995 to 2005) due to the improvements in standards of living (*Tkachenko et al., 2007*). With the development of economic growth in China within decades, the environmental and hygienic conditions were dramatically improved, due to which the prevalence of H. *pylori* infection is decreasing in China (*Nagy, Johansson & Molloy-Bland, 2016*). In consistent with it, the annual positive rates decreased during eight-year period (from 2007 to 2014) in our study (Fig. 1). The age-dependent manner of H. *pylori* positive rate in children may also reflect the inverse relation to the socioeconomic status, sanitation and living conditions in China (*Zhang et al., 2009a*). The increase of prevalence might be the effect of accumulation because that the acquisition rates were higher than the loss rates (*Ozen, Ertem & Pehlivanoglu, 2006*). With the growing of age, expanding range of activity, collective living and meal in high school lead to the increase of exposure to H. *pylori* infection and opportunities to cross infection (*Zhang & Li, 2012*). But it needs to be further investigated.

It was reported that the male predominance of H. *pylori* infection in adults was a global and homogeneous phenomenon, but such predominance was not apparent in children (*De Martel & Parsonnet, 2006*; *Tkachenko et al., 2007*). But our data showed a higher prevalence in boys than girls and in different years age group (Table 2). It is consistent with the study in Brazil that male gender was one of the risk factors for the acquisition and maintenance of the H. *pylori* infection (*Queiroz et al., 2012*). The prevalence of H. *pylori* infection in a community is related to three factors: the incidence rate of infection, the rate of infection loss (either spontaneous eradication or curative treatment) and the relative survival of those with and without infection. Differential incidence, differential antibiotic exposure or differential protective immunity between genders, which lead to greater loss of infection (or seroreversion) in girls or adults women than in men, may explain the different results observed between children and adult studies (*De Martel & Parsonnet, 2006*). On the other hand, it may be explained that boys are naturally more active and have poor personal hygiene than girls as the prevalence of H. *pylori* infection is inversely related to sanitation condition. But the role of gender as a risk factor for H. *pylori* infection is still debated.

Abdominal complaints such as pain, anorexia, nausea/vomiting, or other dyspeptic symptoms are nonspecific and can be caused by different organic disease within and outside the digestive tract. The European Pediatric Task Force concluded in their guidelines on management of H. *pylori* infection that, in children, H. *pylori* infection is not related to gastrointestinal symptoms (*Drumm, Koletzko & Oderda, 2000*). Studies comparing the prevalence among symptomatic and asymptomatic children show different results on the relationship between gastrointestinal symptoms and the prevalence of H. *pylori* infection (*Daugule et al., 2007*; *Dore et al., 2012*). A meta-analysis reported recently that

children with upper abdominal pain or epigastric pain were at two- to three fold higher risk for H. *pylori* infection than children without these symptoms but it could not been confirmed in children seen in primary care (*Spee et al., 2010*). According to multivariate logistic regression analysis, our study showed that gastrointestinal symptom and a history of H. *pylori* infected family member were also the significant risk factors for H. *pylori* infection. Similarly, other studies showed that upper GIT symptoms (RAP, anorexia, nausea), family history of peptic disease, and nausea/vomiting were significantly associated with H. *pylori* infection (*Dore et al., 2012*; *Habib et al., 2014*). However, there are many other possible risk factors associated with H. *pylori* infection identified in most of the published studies, including socioeconomic indicators, family income, household crowding, number of children sharing the same room, parents' education and sharing a bed with children (*Ertem, 2013*). Our results were limited because of cohort selection and the lack of data in these matters, and the determinants of H. *pylori* infection should be investigated by further studies.

In conclusion, the strength of our study was that it evaluated a large number of children in a long period in Hangzhou, a representative city of eastern China. The prevalence of H. *pylori* infection using $^{13}$C-UBT increased with age in children and boys were apt to be H. *pylori* positive compared with girls. The founding suggests that primary infection in childhood is usual and the effect of accumulation might be responsible for the increase of prevalence with age. Besides age and male predominance, gastrointestinal symptom and a history of H. *pylori* infected family member were also the possible risk factors for H. *pylori* infection. In children with history of H. *pylori* infected family member, testing for H. *pylori* may be considered especially when they are symptomatic. These observations could substantially change H. *pylori* management strategies in children in China.

## ACKNOWLEDGEMENTS

We sincerely thank the children and their parents for providing the information to take part in this study. We also thank Lejing Yang and Qian Shu for typewriting the data and thank Kewen Jiang, Weifen Zhu and Xi Chen for suggestions on article editing.

### Funding

This study was supported by a grant from the National Natural Science Foundation of China (No. 81100268). The funders had no role in study design, data collection and analysis, decision to publish, or preparation of the manuscript.

### Grant Disclosures

The following grant information was disclosed by the authors:
National Natural Science Foundation of China: 81100268.

### Competing Interests

The authors declare there are no competing interests.

## Author Contributions

- Xiaoli Shu performed the experiments, analyzed the data, wrote the paper, prepared figures and/or tables.
- Mingfang Ping and Guofeng Yin contributed reagents/materials/analysis tools.
- Mizu Jiang conceived and designed the experiments, reviewed drafts of the paper.

## Human Ethics

The following information was supplied relating to ethical approvals (i.e., approving body and any reference numbers):

The study was approved by the parents and the ethics committees of the Children's hospital, Zhejiang University School of Medicine (2016-IRBAL-078).

## Data Availability

The raw data has been supplied as a Supplemental File.

## Supplemental Information

Supplemental information for this article can be found online at http://dx.doi.org/10.7717/peerj.2937#supplemental-information.

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
