# Peer review of "Investigation of Helicobacter pylori infection among symptomatic children in Hangzhou from 2007 to 2014: a retrospective study with 12,796 cases"

_PeerJ, doi:10.7717/peerj.2937_

## Round 0.1 · original submission · Major Revisions

Dear Xiaoli,

As you see that reviewers have some serious concerns and on balance, they recommend for major revision. Please address all those changes before the manuscript could be considered for acceptance. In particular, the concerns of Reviewer 1 must be appropriately addressed

Best regards,

Ravi Tandon

Reviewer 1 ·

Basic reporting

Manuscript entitled “Low prevalence of Helicobacter pylori infection among symptomatic children in Hangzhou, China” by Shu et al. is evaluating incidence of H. pylori infection among pediatric subjects in province of china that show gastrointestinal symptoms. Such demographic studies are of high importance since these may provide basis for deriving new hypothesis for experiment based investigations towards understanding the disease progression in humans as well as may provide human relevance to evidences collected from studies in other model organisms and in vitro experimental systems. However, these studies require substantial data collection for analyzing correlations among various characteristics of affected individuals as well as require appropriate methods of analysis. Also interpretations derived from the analysis need to be very critically analyze as these may determine future course of investigations. Although the data provided in this manuscript is substantial, the analytical methods and interpretations are questionable and are the major reason why this reviewer find this study unsuitable for publication in the journal “PeerJ”.
The language throughout the manuscript is casual. This reviewer advice to the author for making every effort to use scientific/technical terms as well as refrain from using spoken language while preparing the manuscript for publication. Discussion at few instances is overly speculative and based on author's assumptions e.g., when cause of disparity in H. pylori incidence among individuals of different age groups/genders is discussed.

Experimental design

Major comments:
1. In Section. 1 of the results where demographic data is described, the median should be used rather than mean when age of the individuals in the study is analyzed.
2. A strange analytical method is used throughout data analysis, which makes interpretations very circumstantial if not ultimately wrong. This reviewer recommends analyzing the incidences among any age group/gender as percent of total subjects investigated under this study rather than as percent of the sub-group. This can lead to misinterpretations when subgroups are not equally represented or highly skewed from general tendency. Two striking examples are where authors make case of high frequency of H. pylori cases among individuals with increasing age or with family history of H. pylori infections while data actually suggests the opposite when absolute frequency is estimated. There is strikingly lower frequency of H pylori infection detected in ag group 12-17 whereas the infections are commonly seen in individuals with no family history. Another striking example is where author find no correlation among symptoms and H. pylori infection where in fact in excess of 80% of infected individuals experienced symptoms whereas H. pylori infected individuals very rarely remained asymptomatic (less than 20%). Such misinterpretations will also be reflected in the discussion as the actual data remain underneath the elusive and incorrect conclusions and inappropriate discussion takes place.

Validity of the findings

Due to the lacking described in experimental design, the findings remain erroneous and therefore cannot be validated.

Reviewer 2 ·

Basic reporting

The article does not carry professional standards of courtesy and expression that includes rampant use of language as if in a spoken form (eg: Far below 10%, What’s more, the positive rates were higher in boys than girls, Boys are mischievous than girls")

Experimental design

The reviewer has major concern about the manuscript as the original primary research report.

Validity of the findings

The significance observed in H. pylori infections in boys is relevant but the authors only discussed about socio-economic spectrum of the patients without showing any data.

Additional comments

The observations made by the authors based on the large data sets showed significant gender based difference but the overall data presentation and discussion lacks clarity to engage the reader towards the significance of this observation. Author's previous reports on H.pylori summarizes the epidemiological analysis of this bacterial distribution within the patients.

Reviewer 3 ·

Basic reporting

The manuscript is well written and structured with a standard format.

Experimental design

Experimental design in this study is well organized with sufficient information.

Validity of the findings

The following major points should be revised.
1. This study is based on patients who were referred to H. pylori examination presumably based on physician’s judgement, but not including truly asymptomatic healthy subjects or total pediatric population living in the area. Thus, “prevalence” frequently used in the manuscript including the title potentially gives confusion for the readers, and the authors should consider alternative appropriate terminology. The authors should also discuss the effect of selection bias in this study.

2. The authors showed no significant correlation between gastrointestinal symptoms and H. pylori infection rate. However, according to Table 4, presence of H. pylori family history is associated with high infection rate, so the authors should stratify the comparison of symptomatic vs asymptomatic groups by existence of family history in order to exclude poteial cofounding effect. Actually, given that presence of family history can motivate physicians for referring to H. pylori test even without gastrointestinal symptoms, thus asymptomatic cohort could contain higher rate of positive family history. Particularly, comparison of symptomatic vs asymptomatic within negative family history subjects would greatly support the authors’ conclusions.

3. In order to show a lack of correlation between symptoms and infection, the authors are encouraged to analyze infection rate among groups stratified by the number of symptoms such as no, one, two and more than two symptoms. If there is no elevating tendency depending on the number of symptoms, this data further supports the authors’ conclusions.

4. In Table 4, the authors should indicate odds ratio in similar to Table 3. If the odds ratio is close to one, the authors should carefully discuss the interpretation of this data regardless of statistical significance.

5. In the discussion part (line 178-180), the authors stated that differential infectious rates between male and female could be derived from gender-based characteristics. The authors should provide evidence based on social behavioral analysis, specific to association between H. pylori infection rate and gender-based behavioral factors. Such a speculation for gender characteristics without scientific standard could be an ethic issue. The authors should reorganize this part.

Additional comments

The authors showed low regional infectious rate of H. pylori among tested pediatric population, based on a large-scale cohort including 12796 subjects in China. This study provides precious evidence contributing to comprehension of variability of H. pylori infectious rate in the context of socioeconomic and demographic factors. However, in order to support the main conclusions in this study, there are major issues to be revised as above.

---

## Round 0.2 · accepted · Accept

Dear Xiaoli,
I am pleased to inform that your manuscript has been accepted for the publication after the major revision as proposed by the reviewers.

Reviewer 1 ·

Basic reporting

The authors have made very good effort overall in improving the manuscript.

Experimental design

The multivariate analysis is indeed a very important inclusion and this reviewer admire authors commitment in improving the analytical method and statistics.

Validity of the findings

The data provided in this manuscript is substantial and valuable for understanding the prevalence of helicobacter infections among children. Due to the nature of the study and improved analytical methods in revised manuscripts these findings will be valuable for teh readers.

Reviewer 3 ·

Basic reporting

The manuscript is well written and structured with a standard format.

Experimental design

Experimental design in this study is well organized with sufficient information.

Validity of the findings

Following revisions, conclusions in this manuscript have been supported.

Additional comments

The authors have adequately addressed my comments raised in a previous round of review.